# IS THE PERFORMANCE OF MY DEEP NETWORK TOO GOOD TO BE TRUE? A DIRECT APPROACH TO ESTIMATING THE BAYES ERROR IN BINARY CLASSIFICATION

**Takashi Ishida**[1,2]    **Ikko Yamane**[3]    **Nontawat Charoenphakdee**[1] *
**Gang Niu**[2]    **Masashi Sugiyama**[2,1]
[1]The University of Tokyo    [2]RIKEN    [3]ENSAI/CREST

## ABSTRACT

There is a fundamental limitation in the prediction performance that a machine learning model can achieve due to the inevitable uncertainty of the prediction target. In classification problems, this can be characterized by the Bayes error, which is the best achievable error with any classifier. The Bayes error can be used as a criterion to evaluate classifiers with state-of-the-art performance and can be used to detect test set overfitting. We propose a simple and direct Bayes error estimator, where we just take the mean of the labels that show *uncertainty* of the class assignments. Our flexible approach enables us to perform Bayes error estimation even for weakly supervised data. In contrast to others, our method is model-free and even instance-free. Moreover, it has no hyperparameters and gives a more accurate estimate of the Bayes error than several baselines empirically. Experiments using our method suggest that recently proposed deep networks such as the Vision Transformer may have reached, or is about to reach, the Bayes error for benchmark datasets. Finally, we discuss how we can study the inherent difficulty of the acceptance/rejection decision for scientific articles, by estimating the Bayes error of the ICLR papers from 2017 to 2023.

## 1 INTRODUCTION

Comparing the prediction performance of a deep neural network with the state-of-the-art (SOTA) performance is a common approach to validating advances brought by a new proposal in machine learning research. By definition, SOTA performance such as error or $1-\text{AUC}$[1] monotonically decreases over time, but there is a fundamental limitation in the prediction performance that a machine learning model can achieve. Hence, it is important to figure out how close the current SOTA performance is to the underlying best performance achievable (Theisen et al., 2021). For example, Henighan et al. (2020) studied the scaling law (Kaplan et al., 2020) of Transformers (Vaswani et al., 2017) by distinguishing the *reducible* and *irreducible* errors.

In classification problems, one way to characterize the irreducible part is the *Bayes error* (Cover & Hart, 1967; Fukunaga, 1990), which is the best achievable expected error with any measurable function. The Bayes error will become zero in the special case where class distributions are completely separable, but in practice, we usually deal with complex distributions with some class overlap. Natural images tend to have a lower Bayes error (e.g., 0.21% test error on MNIST (LeCun et al., 1998) in Wan et al. (2013)), while medical images tend to have a higher Bayes error, since even medical technologists can disagree on the ground truth label (Sasada et al., 2018). If the current model's performance has already achieved the Bayes error, it is meaningless to aim for further error improvement. We may even find out that the model's error has *exceeded* the Bayes error. This may imply *test set overfitting* (Recht et al., 2018) is taking place. Knowing the Bayes error will be helpful to avoid such common pitfalls.

Estimating the Bayes error has been a topic of interest in the research community (Fukunaga & Hostetler, 1975; Fukunaga, 1990; Devijver, 1985; Berisha et al., 2016; Noshad et al., 2019; Michelucci

---

*Currently at Preferred Networks.
[1]AUC stands for "Area under the ROC Curve."

et al., 2021; Theisen et al., 2021). To the best of our knowledge, all previous papers have proposed ways to estimate the Bayes error from a dataset consisting of pairs of instances and their hard labels. When instances and hard labels are available, one can also train a supervised classifier, which is known to approach the Bayes classifier (that achieves the Bayes error) with sufficient training data provided that the model is correctly specified.

This is an interesting research problem from the point of view of *Vapnik's principle*[2] (Vapnik, 2000) since we can derive the Bayes error from the Bayes classifier (and the underlying distribution) while we cannot recover the Bayes classifier from the knowledge of the Bayes error, which is just a scalar. How can we take full advantage of this property? While the Bayes error is usually defined as the best achievable expected error with any measurable function, it is known to be equivalent to the expectation of the minimum of class-posteriors with respect to classes for binary classification. Inspired by Vapnik's principle, our main idea is to skip the intermediate step of learning a function model, and we directly approximate the minimum of the class-posteriors by using soft labels (corresponding to the class probability) or uncertainty labels (corresponding to the class uncertainty). [3]

Our proposed method has two benefits. Firstly, our method is *model-free*. Since we do not learn a model, we can escape the curse of dimensionality, while dealing with high-dimensional instances would cause issues such as overfitting if we were to train a model. High dimensionality may cause performance deterioration for other Bayes error estimation methods (Berisha et al., 2016; Noshad et al., 2019) due to divergence estimation. We experimentally show how our method can more accurately estimate the Bayes error than baselines that utilize instances and soft labels. Our model-free method is also extremely fast since we do not have any hyperparameters to tune nor a function model to train.

The second benefit is a more practical one: our method is completely *instance-free*. Suppose our final goal is to estimate the Bayes error instead of training a classifier. In that case, we do not need to collect instance-label pairs, and it may be less costly to collect soft/uncertainty labels without instances. Dealing with instances can cause privacy issues, and it can be expensive due to data storage costs especially when they are high-dimensional or can come in large quantities. It may lead to security costs to protect instances from a data breach. As an example of an instance-free scenario, we can consider doctors who are diagnosing patients by inspecting symptoms and asking questions, without explicitly collecting or storing the patients' data in the database. In this scenario, the hospital will only have the decisions and confidence of doctors, which can be used as soft labels.

The contributions of the paper is as follows. We first propose a direct way to estimate the Bayes error from soft (or uncertainty) labels without a model nor instances. We show that our estimator is unbiased and consistent. In practice, collecting soft/uncertainty labels can be difficult since the labelling process can become noisy. We propose a modified estimator that is still unbiased and consistent even when the soft labels are contaminated with zero-mean noise. We also show that our approach can be applied to other classification problems, such as weakly supervised learning (Sugiyama et al., 2022). Finally, we show the proposed methods' behavior through various experiments. Our results suggest that recently proposed deep networks such as the Vision Transformer (Dosovitskiy et al., 2021) has reached or is about to reach the Bayes error for benchmark datasets, such as CIFAR-10H (Peterson et al., 2019) and Fashion-MNIST-H (which is a new dataset we present; explained in Sec. 5.3). We also demonstrate how our proposed method can be used to estimate the Bayes error for academic conferences such as ICLR, by regarding them as an accept/reject binary classification problem.

## 2 BACKGROUND

Before we discuss our setup, we will review ordinary binary classification from positive and negative data and binary classification from positive-confidence data in this section.

**Learning from positive-negative data**   Suppose a pair of $d$-dimensional instance $\boldsymbol{x} \in \mathbb{R}^d$ and its class label $y \in \{+1, -1\}$ follows an unknown probability distribution with probability density

---

[2]The Vapnik's principle is known as the following: "When solving a given problem, try to avoid solving a more general problem as an intermediate step." (Sec. 1.2 of Vapnik, 2000)

[3]An uncertainty label is 1 subtracted by the dominant class posterior. We will discuss how to obtain soft labels in practice in Sec. 3.

function $p(\boldsymbol{x}, y)$. The goal of binary classification is to train a binary classifier $g \colon \mathbb{R}^d \to \mathbb{R}$ so that the following classification risk $R(g)$ will be minimized: $R(g) = \mathbb{E}_{p(\boldsymbol{x},y)}[\ell(yg(\boldsymbol{x}))]$, where $\ell$ is a margin-based surrogate loss function such as the logistic loss $\ell_{\mathrm{L}}(z) = \log(1 + \exp(-z))$ or the squared loss $\ell_{\mathrm{S}}(z) = (z - 1)^2$. The classification error is $R_{01}(g) = \mathbb{E}_{p(\boldsymbol{x},y)}[\ell_{01}(yg(\boldsymbol{x}))]$, where we use the zero-one loss $\ell_{01}$, which takes 0 when $y \cdot g(\boldsymbol{x}) > 0$ and 1 otherwise. In empirical risk minimization, the following estimator of $R(g)$ is often used:

$$\hat{R}(g) = \tfrac{1}{n} \sum_{i=1}^{n} \ell(y_i g(\boldsymbol{x}_i)), \tag{1}$$

where $\{(\boldsymbol{x}_i, y_i)\}_{i=1}^{n} \overset{\text{i.i.d.}}{\sim} p(\boldsymbol{x}, y)$.

In this paper, our focus is not on the classifier itself but on the value of $R_{01}(g)$ when a good classifier is learned. This can be nicely captured by the notion of the Bayes error, defined as follows:

**Definition 2.1.** (Mohri et al., 2018) The Bayes error is the infimum of the classification risk with respect to all measurable functions: $\beta = \inf_{g \colon \mathbb{R}^d \to \mathbb{R}, \text{ measurable}} \mathbb{E}_{p(\boldsymbol{x},y)}[\ell_{01}(yg(\boldsymbol{x}))]$.

**Learning from positive-confidence data** In positive-confidence classification (Ishida et al., 2018; Shinoda et al., 2020), we only have access to positive data equipped with confidence $\{(\boldsymbol{x}_i, r_i)\}_{i=1}^{n_+}$, where $\boldsymbol{x}_i$ is a positive instance drawn independently from $p(\boldsymbol{x} \mid y = +1)$ and $r_i$ is the positive confidence given by $r_i = p(y = +1|\boldsymbol{x}_i)$. Since we only have positive instances, we cannot directly minimize Eq. (1). However, Ishida et al. (2018) showed that the following theorem holds.

**Theorem 2.2.** *(Ishida et al., 2018) The classification risk can be expressed as*

$$R(g) = \pi_+ \mathbb{E}_+ \left[ \ell(g(\boldsymbol{x})) + \tfrac{1 - r(\boldsymbol{x})}{r(\boldsymbol{x})} \ell(-g(\boldsymbol{x})) \right] \tag{2}$$

*if we have $r(\boldsymbol{x}) = p(y = +1|\boldsymbol{x}) \neq 0$ for all $\boldsymbol{x}$ sampled from $p(\boldsymbol{x})$. $\mathbb{E}_+$ is the expectation over the positive density $p(\boldsymbol{x}|y = +1)$ and $\pi_+ = p(y = +1)$.*

Since the expectation is over $p(\boldsymbol{x}|y = +1)$, we do not have to prepare negative data. Although $\pi_+$ is usually unknown, we do not need this knowledge since $\pi_+$ can be regarded as a proportional constant when minimizing Eq. (2). Hence, we can see how learning a classifier is possible in this weakly-supervised setup, but is estimating the Bayes error also possible? We will study this problem further in Sec. 4.

## 3 Estimation with ordinary data

In this section, we propose our direct approach to Bayes error estimation.

**Clean soft labels** First of all, it is known that the Bayes error can be rewritten as follows (Devroye et al., 1996; Mohri et al., 2018):

$$\beta = \mathbb{E}_{\boldsymbol{x} \sim p(\boldsymbol{x})} \left[ \min\{ p(y = +1 \mid \boldsymbol{x}), p(y = -1 \mid \boldsymbol{x}) \} \right]. \tag{3}$$

This means that we can express the Bayes error $\beta$ without explicitly involving the classifier $g$. If we assume that we have soft labels, which are $\{c_i\}_{i=1}^{n}$, where $c_i = p(y = +1 \mid \boldsymbol{x}_i)$ and $\{\boldsymbol{x}_i\}_{i=1}^{n} \overset{\text{i.i.d.}}{\sim} p(\boldsymbol{x})$, we can construct an extremely straightforward estimator for the Bayes error:

$$\hat{\beta} = \tfrac{1}{n} \sum_{i=1}^{n} \min\{c_i, 1 - c_i\}. \tag{4}$$

In fact, Eq. (4) is just the empirical version of Eq. (3). However, the significance lies in the observation that this can be performed in a *model-free* and *instance-free* fashion, since taking the averages of the smaller one out of $c_i$ and $1 - c_i$ in Eq. (4) does not require model training and does not involve $\boldsymbol{x}_i$.

Note that we can further weaken the supervision by sampling *uncertainty labels*, i.e., $c_i' = \min_{y \in \{\pm 1\}} p(y \mid \boldsymbol{x}_i)$. Practically, this would mean that the label provider does not need to reveal the soft label. Although the two are similar, we can see how uncertainty labels are weaker by the fact that we cannot recover $p(y = +1 \mid \boldsymbol{x})$ from $\min_{y \in \{\pm 1\}} p(y \mid \boldsymbol{x})$[4]. Furthermore, we can recover $\pi_+ = p(y = +1)$ by the expectation of soft labels $\mathbb{E}_{p(\boldsymbol{x})}[p(y = +1 \mid \boldsymbol{x})] = \pi_+$, but $\pi_+$ cannot be recovered by averaging uncertainty labels.

Our estimator enjoys the following theoretical properties.

---

[4]For example, if $\min_{y \in \{\pm 1\}} p(y \mid \boldsymbol{x}) = 0.1$, we do not know whether $p(y = +1 \mid \boldsymbol{x})$ is 0.9 or 0.1.

**Proposition 3.1.** $\hat{\beta}$ *is an unbiased estimator of the Bayes error.*

**Proposition 3.2.** *For any $\delta > 0$, with probability at least $1 - \delta$, we have, $|\hat{\beta} - \beta| \leq \sqrt{\frac{1}{8n} \log \frac{2}{\delta}}$.*

The proofs are shown in App. C.1 and App. C.2, respectively. This implies that $\hat{\beta}$ is a consistent estimator to the Bayes error, i.e., $\forall \epsilon > 0$, $\lim_{n \to \infty} P(|\hat{\beta} - \beta| \geq \epsilon) = 0$. The rate of convergence is $|\hat{\beta} - \beta| = \mathcal{O}_P(1/\sqrt{n})$, where $\mathcal{O}_P$ is the stochastic big-O notation, cf. App. D. This rate is the same as classifier training and is the optimal parametric rate (Mendelson, 2008) although our method can be regarded as non-parametric. We are also free from the curse of dimensionality since the instance dimension $d$ does not appear in the bound.

**Noisy soft labels** So far, our discussions were based on an assumption that soft (or uncertainty) labels is precisely the class-posterior $c_i$ (or $c_i'$). Since this may not be satisfied in practice, we consider a situation where noise $\xi_i$ is added to the class-posterior values. We observe noisy soft labels $\{u_i\}_{i=1}^n$ where $u_i = c_i + \xi_i$. Since we ask labellers to give us a probability, a constraint is $0 \leq u_i \leq 1$, and we assume $\mathbb{E}[\xi_i | c_i] = 0$. If we naively use these noisy versions, our estimator becomes

$$\tilde{\beta} = \frac{1}{n} \sum_{i=1}^n \left[ \min \left( u_i, 1 - u_i \right) \right]. \tag{5}$$

However, we can show the following negative result.

**Theorem 3.3.** $\tilde{\beta}$ *is generally a* biased *estimator of the Bayes error:* $\mathbb{E}_{\{(c_i, \xi_i)\}_{i=1}^n \overset{\text{i.i.d.}}{\sim} p(c, \xi)}[\tilde{\beta}] \neq \beta$.

This is basically due to the noise inside the nonlinear $\min$ operation. We show the proof in App. C.3.

This issue can be fixed if we have an additional binary signal that shows if the instance is likely to be more positive or negative. Formally, if we have access to sign labels, i.e., $s_i = \text{sign}\left[ p(y = +1 | \boldsymbol{x}_i) - 0.5 \right] \quad \forall i \in \{1, 2, \ldots, n\}$, we can divide $\{u_i\}_{i=1}^n$ into $\{u_i^+\}_{i=1}^{n_s^+}$ and $\{u_i^-\}_{i=1}^{n_s^-}$ depending on the sign label, with $n = n_s^+ + n_s^-$. Then we can construct the following estimator:

$$\hat{\beta}_{\text{Noise}} = \frac{1}{n} \left( \sum_{i=1}^{n_s^+} \left( 1 - u_i^+ \right) + \sum_{i=1}^{n_s^-} u_i^- \right). \tag{6}$$

With this modified estimator $\hat{\beta}_{\text{Noise}}$, we can show the following propositions.

**Proposition 3.4.** $\hat{\beta}_{\text{Noise}}$ *is an unbiased estimator of the Bayes error.*

The proof is shown in App. C.5. The proof itself is simple, but the main idea lies in where we modified the estimator from Eq. (4) to avoid dealing with the noise inside the $\min$ operator, by utilizing the additional information of $s_i$.

**Proposition 3.5.** *For any $\delta > 0$, with probability at least $1 - \delta$, we have, $|\hat{\beta}_{\text{Noise}} - \beta| \leq \sqrt{\frac{1}{2n} \log \frac{2}{\delta}}$.*

The proof is shown in App. C.6. This implies that $\hat{\beta}_{\text{Noise}}$ is a consistent estimator to the Bayes error, i.e., for $\forall \epsilon > 0$, $\lim_{n \to \infty} P(|\hat{\beta}_{\text{Noise}} - \beta| \geq \epsilon) = 0$. The rate of convergence is $\mathcal{O}_P(1/\sqrt{n})$.

In practice, one idea of obtaining soft labels is to ask many labellers to give a hard label for each instance and then use the proportion of the hard labels as a noisy soft label. This is the main idea which we use later on in Sec. 5.2 with CIFAR-10H (Peterson et al., 2019). For simplicity, suppose that we have $m$ labels for every instance $i$, and $u_i$ is the sample average of those labels: $u_i = \frac{1}{m} \sum_{j=1}^m \mathbb{1}[y_{i,j} = 1]$, where $y_{i,j}$ is the $j$th hard label for the $i$th instance and $\mathbb{1}$ denotes the indicator function. This satisfies our assumptions for noisy soft labels since $0 \leq u_i \leq 1$ and $\mathbb{E}[\xi_i | c_i] = 0$ for any $m \in \mathbb{N}$. If in practice we do not have access to sign labels, we cannot use the unbiased estimator $\hat{\beta}_{\text{Noise}}$ and we have no choice but to use the biased estimator $\tilde{\beta}$. However, in the following proposition, we show that the bias can be bounded with respect to $m$.

**Proposition 3.6.** *The estimator $\tilde{\beta}$ is asymptotically unbiased, i.e., $|\beta - \mathbb{E}[\tilde{\beta}]| \leq \frac{1}{2\sqrt{m}} + \sqrt{\log(2n\sqrt{m})/(2m)}$, where $m$ is the number of labellers for each of the $n$ instances.*

The proof is shown in App. C.4. The rate of convergence is $\mathcal{O}(\sqrt{\log(2nm)/m})$.

**Hard labels**  Surprisingly, we can regard the usual hard labels as a special case of noisy soft labels. This enables us to estimate the Bayes error with hard labels and sign labels. To see why this is the case, we consider a specific type of noise: $\xi_i$ is $1 - p(y = +1 \mid \boldsymbol{x}_i)$ w.p. $p(y = +1 \mid \boldsymbol{x}_i)$ and $-p(y = +1 \mid \boldsymbol{x}_i)$ w.p. $1 - p(y = +1 \mid \boldsymbol{x}_i)$. Since $u_i = c_i + \xi_i$, this implies $u_i = 1$ w.p. $p(y = +1 | \boldsymbol{x}_i)$ and $u_i = 0$ w.p. $p(y = -1 | \boldsymbol{x}_i)$. We can easily confirm that our assumptions for noisy soft labels are satisfied: $0 \leq \xi_i \leq 1$ and $\mathbb{E}[\xi_i \mid c_i] = 0$.

If we assign $y = +1$ for $u_i = 1$ and $y = -1$ for $u_i = 0$, this will be equivalent to providing a hard label based on a Bernoulli distribution $\mathrm{Be}(p(y = +1 | \boldsymbol{x}_i))$ for an instance $\boldsymbol{x}_i$ sampled from $p(\boldsymbol{x})$. This corresponds to the usual data generating process in supervised learning.

## 4  ESTIMATION WITH POSITIVE-CONFIDENCE DATA

In this section, we move our focus to the problem of estimating the Bayes error with positive-confidence (Pconf) data (Sec. 2). First, we show the following theorem.

**Theorem 4.1.** *The Bayes error can be expressed with positive-confidence data and class prior:*

$$\beta_{\mathrm{Pconf}} := \pi_+ \left( 1 - \mathbb{E}_+ \left[ \max \left( 0, 2 - \tfrac{1}{r(\boldsymbol{x})} \right) \right] \right) = \beta. \tag{7}$$

Recall that $\mathbb{E}_+$ is $\mathbb{E}_{p(\boldsymbol{x}|y=+1)}$ and $r(\boldsymbol{x}) = p(y = +1 | \boldsymbol{x})$. The proof is shown in App. C.7. Based on Theorem 4.1, we can derive the following estimator:

$$\hat{\beta}_{\mathrm{Pconf}} = \pi_+ \left( 1 - \tfrac{1}{n_+} \sum_{i=1}^{n_+} \max \left( 0, 2 - \tfrac{1}{r_i} \right) \right), \tag{8}$$

where $r_i = p(y = +1 | \boldsymbol{x}_i)$ for $\boldsymbol{x}_i \sim p(\boldsymbol{x}|y = +1)$, and $n_+$ is the number of Pconf data. We can estimate the Bayes error in an instance-free fashion, even if we do not have any $r_i$ from the negative distribution. All we require is instance-free positive data and $\pi_+$. A counter-intuitive point of Eq. (8) is that we can throw away $r_i$ when $r_i < 0.5$, due to the $\max$ operation. If the labeler is confident that $r_i < 0.5$, we do not need the specific value of $r_i$. Note that in practice, $r_i = 0$ may occur for positive samples. This will cause an issue for classifier training (since Eq. (2) will diverge), but such samples can be ignored in Bayes error estimation (due to the $\max$ in Eq. (8)).

As discussed in Sec. 2, $\pi_+$ was unnecessary when learning a Pconf classifier, but it is necessary for Bayes error estimation. Fortunately, this is available from other sources depending on the task, e.g., market share, government statistics, or other domain knowledge (Quadrianto et al., 2009; Patrini et al., 2014). Instead of the Bayes error itself, we may be interested in the following: 1) the relative comparison between the Bayes error and the classifier's error, 2) whether the classifier's error has exceeded the Bayes error or not. In these cases, $\pi_+$ is unnecessary since we can divide both errors (Eq. (8) and empirical version of Eq. (2)) by $\pi_+$ without changing the order of the two.

We can derive the following two propositions.

**Proposition 4.2.** $\hat{\beta}_{\mathrm{Pconf}}$ *is an unbiased estimator of the Bayes error.*

**Proposition 4.3.** *For any $\delta > 0$, with probability at least $1 - \delta$, we have, $|\hat{\beta}_{\mathrm{Pconf}} - \beta| \leq \sqrt{\frac{\pi_+^2}{2n_+} \log \frac{2}{\delta}}$.*

The proofs are shown in App. C.8 and App. C.9, respectively. This implies that the estimator $\hat{\beta}_{\mathrm{Pconf}}$ is consistent to the Bayes error, i.e., for $\forall \epsilon > 0$, $\lim_{n \to \infty} P(|\hat{\beta}_{\mathrm{Pconf}} - \beta| \geq \epsilon) = 0$. The rate of convergence is $\mathcal{O}_P(1/\sqrt{n_+})$. We can also see that the upper bound in Prop. 4.3 is equivalent to the upper bound in Prop. 3.2 when the data is balanced, i.e., $\pi_+ = 0.5$.

## 5  EXPERIMENTS

In this section, we first perform synthetic experiments to investigate the behavior of our methods. Then, we use benchmark datasets to compare the estimated Bayes error with SOTA classifiers. Finally, we show a real-world application by estimating the Bayes error for ICLR papers. [5]

---

[5]Our code will be available at https://github.com/takashiishida/irreducible.

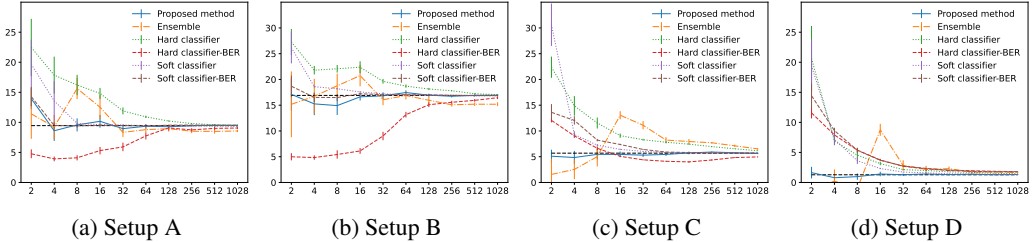

|  |  |  |  |
|---|---|---|---|
| (a) Setup A | (b) Setup B | (c) Setup C | (d) Setup D |

Figure 1: Comparison of the proposed method (blue) with the true Bayes error (black), a recent Bayes error estimator "Ensemble" (orange) (Noshad et al., 2019), and the test error of a multilayer perceptron (hard labels: green, soft labels: purple). We also use the classifier's probabilistic output and plug it into the Bayes error equation in Eq. (3), shown as "Classifier-BER" (hard labels: red, soft labels: brown). The vertical/horizontal axis is the test error or estimated Bayes error/# of training samples per class, respectively. Instance-free data was used for Bayes error estimation, while fully labelled data was used for training classifiers. We plot the mean and standard error for 10 trials.

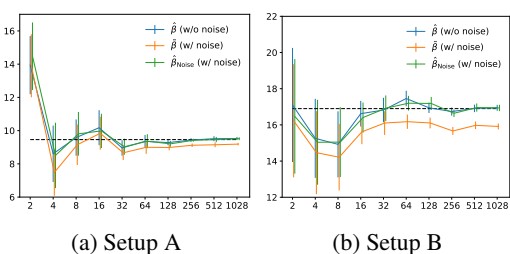

| (a) Setup A | (b) Setup B |
|---|---|

Figure 2: Bayes error with and without noise on the labels, with mean and standard error for 10 trials. The vertical/horizontal axis is the estimated Bayes error/the number of samples. With noisy labels, the estimation tends to become inaccurate with a larger standard error. With more samples, the difference between clean labels becomes smaller, but a small bias remains. With our modified estimator, the bias vanishes.

## 5.1 ILLUSTRATION

**Positive-negative data**   **Setup:** We use two Gaussian distributions to construct a binary classification dataset. We then derive the positive-class posterior analytically for each data and use it as our label. We use Eq. (4) to estimate the Bayes error. We change the number of training samples per class $(2, 4, 8, \ldots, 1028)$ to observe how quickly we can approach the true Bayes error. We perform experiments with four setups (A, B, C, and D) and the details are explained after Sec. 6. To compare with the true Bayes error of the setup, we take the average of $c(\boldsymbol{x})$ for $10{,}000$ samples. We expect this to be a good substitute because we are using the analytically derived class posteriors for the soft labels. Although our problem setting does not require instances, we also compare with a recent Bayes error estimator that uses instances and hard labels (Noshad et al., 2019) and classifiers trained from instances and hard labels. We train a multi-layer perceptron model with two hidden layers (500 units each, followed by BatchNorm (Ioffe & Szegedy, 2015) and ReLU (Nair & Hinton, 2010)). We use the following cross-entropy loss: $-\mathbb{1}[y_i = +1] \log \hat{p}_\theta(y = +1|\boldsymbol{x}_i) - \mathbb{1}[y_i = -1] \log \hat{p}_\theta(y = -1|\boldsymbol{x}_i)$, where $\hat{p}_\theta(y|\boldsymbol{x})$ is the probabilistic output of the trained classifier and $\theta$ represents the model parameters. We chose the cross entropy loss because it is a proper scoring rule (Kull & Flach, 2015). We use stochastic gradient descent for 150 epochs with a learning rate of 0.005 and momentum of 0.9. We also compare with another baseline ("classifier-BER"), where we plug in $\hat{p}_\theta(y|\boldsymbol{x})$ to Eq. (3). Finally, we add a stronger baseline "soft classifier" which utilizes soft labels. If the soft label is $r_i$ for sample $\boldsymbol{x}_i$, then the cross-entropy loss is modified as follows: $-r_i \log \hat{p}_\theta(y = +1|\boldsymbol{x}_i) - (1 - r_i) \log \hat{p}_\theta(y = -1|\boldsymbol{x}_i)$.

**Results:** We construct a dataset with the same underlying distribution ten times and report the means and standard error for the final epoch in Fig. 1. We can see how the estimated Bayes error is relatively accurate even when we have few samples. The benefit against classifier training is clear: the MLP model requires much more samples before the test error approaches the Bayes error. The classifier is slightly better when it can use soft labels but is not as good as our proposed method. It is interesting how the "hard classifier-BER" tends to underestimate the Bayes error (except Fig. 1d). This can be explained by the overconfidence of neural networks (Guo et al., 2017), which will lead to a small $\min_{y \in \{\pm 1\}} \hat{p}(y|\boldsymbol{x})$. This underestimation does not occur with "soft classifier-BER," implying that there are less room to be overconfident when we learn with the modified cross-entropy loss.

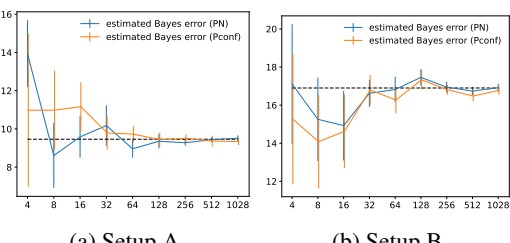

(a) Setup A       (b) Setup B

Figure 3: Estimated Bayes error with PN (positive and negative) and Pconf (positive-confidence). The vertical/horizontal axis is the estimated Bayes error/number of samples. We plot the mean and standard error for 10 trials. The difference between the two becomes smaller as we increase the number of training samples.

**Positive-negative data with noise** **Setup:** Next, we perform experiments when we have noise on the labels. We follow the assumption in Sec. 3, and we design the noise to have mean 0 and the label to be between 0 and 1 after adding this noise, to be a valid probability. For this purpose, we add noise to the underlying class posterior and use a truncated Gaussian distribution $p(z)$ with standard deviation 0.4. We truncate the left tail $\{z \in \mathbb{R} | z \leq \mu - \min(1 - \mu, \mu - 0)\}$ and the right tail $\{z \in \mathbb{R} | z \geq \mu + \min(1 - \mu, \mu - 0)\}$ where $\mu \in [0, 1]$ is the mean of the distribution, i.e., the true class posterior. We compare three estimators: 1) Original estimator $\hat{\beta}$ in Eq. (4) with clean labels, 2) original estimator $\tilde{\beta}$ in Eq. (4) with noisy labels, and 3) proposed estimator $\hat{\beta}_{\text{Noise}}$ in Eq. (6) with noisy labels. We repeat the experiment 10 times with different data realizations. **Results:** The mean and standard error for 10 trials are shown in Fig. 2, for varying number of training samples. As expected, using noisy soft labels lead to worse results for $\tilde{\beta}$, as we can see with the larger standard errors. Increasing the sample size is effective to some degree, but the bias does not vanish. However, if we use a modified estimator $\hat{\beta}_{\text{Noise}}$, the bias vanishes completely and works well even *with* noise.

**Positive-confidence data** **Setup:** Similarly, we perform experiments for positive-confidence (Pconf) data. The data is generated so that the class prior $\pi_+$ is 0.5, and we assume this information is available. We only use the confidence of data from the positive distribution and do not use the confidence of data from the negative distribution. We use this data to estimate $\hat{\beta}_{\text{Pconf}}$ and compare it with the $\hat{\beta}$ derived earlier. **Results:** We report the mean and standard error for 10 trials in Fig. 3. Although Bayes error estimation tends to be inaccurate with Pconf data for fewer samples, the difference between the two disappears with more samples.

## 5.2 EXPERIMENTS WITH BENCHMARK DATASETS: CIFAR-10

**Setup:** We use CIFAR-10 (C10) (Krizhevsky, 2009), ciFAIR-10 (C10F) (Barz & Denzler, 2020), CIFAR-10.1 (C10.1) (Recht et al., 2018), and CIFAR-10H (C10H) (Peterson et al., 2019). C10 is an image dataset of 10 classes: airplane, automobile, bird, cat, deer, dog, frog, horse, ship, and truck. However, the test set in C10 is known to include complete duplicates and near-duplicates from the training dataset. C10F replaces the duplicates in the test set with a different image. C10.1 is a new test set to evaluate C10 classifiers. It ensures the underlying distribution to be as close to the original distribution as possible, but it does not include any images from the original test set. C10H is a dataset with additional human annotations for the C10 test set. Each image in the test set has class labels annotated by around 50 different human labelers. We can take the proportion to construct soft labels that we will use for Bayes error estimation. Although these datasets are originally multi-class setups with 10 classes, we re-construct binary datasets in the following four ways: animals vs. artifacts, land vs. other (water/sky), odd vs. even, and first five vs. last five. See the details after Sec. 6.

We compare the estimated Bayes error $\hat{\beta}$ with the base model of Vision Transformer (Dosovitskiy et al., 2021) with $16 \times 16$ input patch size (ViT B/16), DenseNet169 (Huang et al., 2017), ResNet50 (He et al., 2016), and VGG11 (Simonyan & Zisserman, 2015). For ViT B/16, we downloaded the official pretrained model (Dosovitskiy et al., 2021) and used the implementation by Jeon (2020) to fine-tune on C10. For others, we use the models provided in Phan (2021), which are trained on C10. Although these are ten-class classifiers, we transform them into binary classifiers by swapping the predictions into positive and negative based on the setups (details shown after Sec. 6). For example, a prediction of "cat" would be swapped with a positive label in "animals vs. artifacts".

**Results:** We compare the estimated Bayes error $\hat{\beta}$ with classifier training in Fig. 4. As expected, the test error (shown in light blue) is lower for newer network architectures, with ViT achieving the

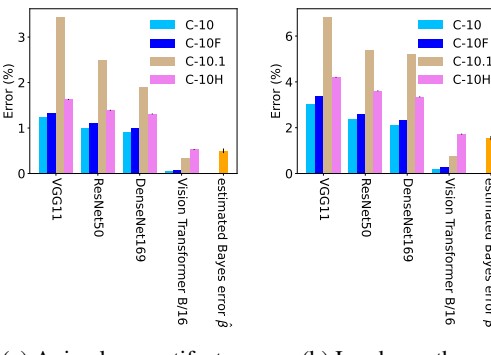

(a) Animals vs. artifacts    (b) Land vs. other

Figure 4: Comparing the estimated Bayes error with four classifiers and two setups (Figures 4a, 4b). Although it seems that ViT B/16 has surpassed the Bayes error for all four setups for CIFAR-10 (C10), ciFAIR-10 (C10F), and CIFAR-10.1 (C10.1), it is still comparable or larger than the Bayes error for CIFAR-10H (C10H). In Sec. 5.2, we discuss how this has to be investigated carefully. The black error bar for the Bayes error is the 95% confidence interval. The black error bar for the C10H is the standard error for sampling test labels 20 times.

lowest test error. An interesting observation is that the test error of ViT has exceeded the estimated Bayes error in all four setups, which theoretically should not take place. We first suspected that ViT's error is too low (overfitting to the test set; Sec. B) but found out this is not necessarily the case. We investigated the degree of overfitting to the test set by performing experiments with C10F and C10.1. The former dataset excludes the duplicated images between training and test set, and the latter dataset is a fresh test set that is designed to be close to the original distribution of C10. The test error for the C10F dataset becomes slightly worse and much worse for C10.1. This is what we expected and implies that overfitting to the test set may be happening to some degree. However, ViT's error is *still* below the estimated Bayes error even for C10.1.

To explain why ViT's error is still below the estimated Bayes error, we discuss the discrepancy between the underlying $p(y|\boldsymbol{x})$ for C10 and C10H. C10 is constructed with a three-step procedure: first, collect images by searching for specific terms in search engines, downscale them to $32 \times 32$ (Torralba et al., 2008), and finally ask labelers to filter out mislabelled images (Krizhevsky, 2009). On the other hand, C10H constructs the labels by asking labelers to choose the correct class out of ten classes after seeing each *downscaled* image of C10. This makes it visually harder for human labellers. Hence, the labelling process will become less confident, and this may lead to a higher entropy of the distribution of the human labels. According to Eq. (4), this will lead to an overestimation of $\hat{\beta}$.

To adjust for the distribution discrepancy, we also plot the test errors when we sample a test label for each sample from the C10H dataset (repeated 20 times) and show this in Fig. 4 with pink. Note that this generally will not go below the estimated Bayes error. For example, consider an extreme case with zero test error for the original test set. If a test sample has $p(y = +1|\boldsymbol{x}) = 0.8$ in C10H, then it will incur an error of 0.2 on average for the new test labels. After adjusting for the discrepancy, we can see how the test error of ViT is on par with or slightly higher than the estimated Bayes error. This implies that we have reached (or is about to reach) the Bayes error for C10H.

## 5.3 EXPERIMENTS WITH BENCHMARK DATASETS: FASHION-MNIST

**Setup:** To our knowledge there is only one large-scale dataset with over 10 annotations per sample (Peterson et al., 2019), which we used in Sec. 5.2. Hence, it would be beneficial to construct another benchmark dataset with multiple annotations. We collected multiple annotations for each test image in Fashion-MNIST (Xiao et al., 2017). Similar to CIFAR-10H, we name our dataset as Fashion-MNIST-H. See our data collection details in App. E.

Since aiming for SOTA performance on Fashion-MNIST is less common in our community compared with CIFAR-10, we simply chose ResNet-18 (He et al., 2016) for our model architecture. We trained for 100 epochs with batch size 128, learning rate 0.001 on the original labels (no data augmentation) and reconstructed the 10-class predictions into binary predictions by grouping shirt, dress, coat, pullover, and t-shirt/top as the positive class, and the other 5 categories as the negative class.

**Results:** We compared the binary predictions with a test label sampled from the multiple annotations (repeated 10 times) and derived the error, which is basically the same procedure of the pink bar in Fig. 4. The error was 3.852% ($\pm$ 0.041%), where the percentage in the parentheses is the standard error. On the other hand, the estimated Bayes error was 3.478% ($\pm$ 0.079%), where the percentage in the parentheses is the 95% confidence interval. We can see how the estimated Bayes error is

higher than the results for CIFAR-10H in Fig. 4, implying that Fashion-MNIST is more difficult than CIFAR-10. This is intuitive as the $28 \times 28$ grayscale images are ambiguous while the original $51 \times 73$ PNG image is much more clear, e.g., see Fig. 1 in Xiao et al. (2017), and also consistent with the results in Theisen et al. (2021). The ResNet-18 error is approaching the estimated Bayes error, but may still have room for small improvement. Since we made a simple choice of model architecture, we may be able to close this gap by other architectures that are more suited for Fashion-MNIST images, adding data augmentation, and trying extensive hyper-parameter tuning.

## 5.4 EXPERIMENTS WITH REAL-WORLD DATASETS: ACADEMIC PAPER CLASSIFICATION

We consider an accept/reject binary classification task for ICLR papers and estimate the Bayes error for this task. We believe this is an interesting real-world application because it may be used as a tool to help us gain a deeper understanding of the trends/characteristics of academic conferences.

**Setup:** We used the OpenReview API to get data from ICLR 2017 to 2023. We removed papers that do not have a decision, e.g., withdrawn or desk-rejected papers. This real-world application is interesting because we have access to expert soft labels, i.e., the recommendation scores of the reviewers. We can derive a weighted average of the scores based on the confidence provided by the reviewer, $(\sum_i c_i \cdot s_i)/(\sum_i c_i)$, where $c_i$ is the confidence and $s_i$ is the score for reviewer $i$, respectively. In ICLR, this average is between 1 and 10 and is not a probability. Fortunately, we also have access to the hard label (final decision) for each paper which we can use for calibration. We used Platt scaling (Platt et al., 1999) with the hard labels so that we have a calibrated score between 0 and 1. Then we use our proposed estimator to derive the Bayes error for each year of ICLR.

**Assumptions of the data generation process:** First, a reviewer provides a score between 0 and 1 for each paper. Then the Area Chair (or SAC/PC) samples a binary label (accept/reject) based on a Bernoulli distribution with the score provided by the reviewers. We feel this is realistic since the AC usually accepts/rejects papers with very high/low average scores while the AC puts more effort into papers with borderline scores meaning that papers close to borderline can end up both ways. These are the papers responsible for raising the Bayes error. Our motivation for deriving Bayes error estimates for conference venues is to investigate this kind of intrinsic difficulty.

| ICLR's Bayes error | |
|---|---|
| 2017 | $6.8\%(\pm 1.0\%)$ |
| 2018 | $8.7\%(\pm 0.9\%)$ |
| 2019 | $7.9\%(\pm 0.7\%)$ |
| 2020 | $8.8\%(\pm 0.5\%)$ |
| 2021 | $9.3\%(\pm 0.5\%)$ |
| 2022 | $9.6\%(\pm 0.5\%)$ |
| 2023 | $8.0\%(\pm 0.4\%)$ |

Table 1: Parenthesis is 95% confidence interval.

**Results:** From Table 1, we can see that the Bayes error is stable to some extent and there are no disruptive changes over the years, e.g., Bayes error is not doubling or tripling. We can also confirm that Bayes error for ICLR paper accept/reject prediction is higher than the other benchmark experiments with CIFAR-10 and Fashion-MNIST, which is expected since understanding the value of a paper is much more difficult than identifying dogs, cars, t-shirts, etc.

This example demonstrates the benefit of our instance-free approach. At least at the time of writing, we do not rely on state-of-the-art ML to review papers submitted to academic conferences/journals. Even if we wanted to, it is not clear what the features would be. Since new papers are reviewed based on the novelty compared to all previous papers, how to handle this is nontrivial. This implies we cannot easily use instance-based Bayes error estimators. It also demonstrates a real-world situation where we can directly sample soft labels from experts instead of collecting multiple hard labels.

## 6 CONCLUSIONS

We proposed a model-free/instance-free Bayes error estimation method that is unbiased and consistent. We showed how we can estimate it with uncertainty labels, noisy soft labels, or positive-confidence. In experiments, we demonstrated how Bayes error estimation can be used to compare with the SOTA performance for benchmark datasets, and how it can be used to study the difficulty of datasets, e.g., paper selection difficulty. Although we did not detect strong evidence of test set overfitting in Secs. 5.2 and 5.3, we are interested in investigating other benchmark datasets in the future.

## ETHICS STATEMENT

Constructing soft labels may rely on collecting human annotations. This may potentially have a negative impact, e.g., cost of the annotations. We are hoping to lower the cost of annotations in our future work. We will only publish our annotations we collected in Sec. 5.3 in a way that does not contain any private information, e.g., only the category counts for each images and drop all other worker-related information.

## REPRODUCIBILITY STATEMENT

We will upload the code for implementing our proposed methods and labels of Fashion-MNIST-H in `https://github.com/takashiishida/irreducible`.

**Synthetic experiments:** In the synthetic experiments, we used four setups: A, B, C, and D. A and B are 2-dimensional anisotropic setups and is borrowed from the first two synthetic datasets in Ishida et al. (2018), where the code is provided in `https://github.com/takashiishida/pconf`. C is a 10-dimensional isotropic setup with means $\boldsymbol{\mu}_+ = [0, \ldots, 0]^\top$ and $\boldsymbol{\mu}_- = [1, \ldots, 1]^\top$ with identity matrices for the covariance matrices. D is the same as C except that the dimension is doubled to 20.

We used `https://github.com/mrtnoshad/Bayes_Error_Estimator` for the ensemble method (Noshad et al., 2019). To make it advantageous for this baseline, we used `KNN` for `bw_selection` when we had 16 or more samples per class but used `manual` when we had less than 16. We partially used `https://github.com/takashiishida/flooding` for training classifiers in synthetic experiments with Gaussian distributions.

**Benchmark experiments:** We used data and code provided in `https://cvjena.github.io/cifair/` for ciFAIR-10 (Barz & Denzler, 2020). We used data and code provided in `https://github.com/jcpeterson/cifar-10h` for CIFAR-10H (Peterson et al., 2019). We used data and code provided in `https://github.com/modestyachts/CIFAR-10.1` for CIFAR-10.1 (Recht et al., 2019) and PyTorch wrapper provided in `https://github.com/kharvd/cifar-10.1-pytorch`. We used data provided in `https://www.cs.toronto.edu/~kriz/cifar.html` for CIFAR-10. We used `https://github.com/huyvnphan/PyTorch_CIFAR10` for VGG11, ResNet50, and DenseNet169 models trained on CIFAR-10. We used the ResNet-18 implementation provided in `https://github.com/rasbt/deeplearning-models`. We used `https://github.com/jeonsworld/ViT-pytorch` to fine-tune ViT and downloaded pretrained weights from `https://console.cloud.google.com/storage/browser/vit_models;tab=objects?pli=1&prefix=&forceOnObjectsSortingFiltering=false`.

We are not aware of any personally identifiable information or offensive content in the datasets we used. We also collected annotations by using Amazon Mechanical Turk (Buhrmester et al., 2016). We explained the details of how we collected the annotations in Sec. 5.3, and we plan to remove all personally identifiable information when we publish the data.

The details of the two (or four) binary datasets that are used in Sec. 5.2 (or App. 8) are as follows.

- Animals vs. artifacts: we divide the dataset into cat, deer, dog, frog, and horse for the positive class and plane, car, ship, truck, and bird for the negative class.
- Land vs. other (water/sky): we divide the dataset into car, truck, cat, deer, dog, horse for the positive class and plane, ship, bird, and frog for the negative class.
- Odd vs. even: we divide the dataset into odd and even class numbers, with plane, bird, deer, frog, and ship for the positive class and car, cat, dog, horse, and truck for the negative class.
- First five vs. last five: we divide the dataset into the first five and last five classes based on the class number, with plane, car, bird, cat, and deer for the positive class and dog, frog, horse, ship, and truck for the negative class.

**Real-world experiments:** We used the OpenReview API (`https://docs.openreview.net/`) and openreview-py (`https://pypi.org/project/openreview-py/`) to collect reviewers' scores for papers from ICLR 2017 to ICLR 2023.

All of our experiments were conducted with a single Mac Studio except when we trained ResNet-18 in Sec. 5.3 and fine-tuned ViT in Sec. 5.2, which was when we utilized Google Colab Pro+ in https://colab.research.google.com.

## ACKNOWLEDGEMENTS

We thank Taiji Suzuki, Hiroshi Kera, Futoshi Futami, and Toshihiro Kamishima for the helpful discussions. TI was supported by JST, ACT-X Grant Number JPMJAX2005, Japan. IY acknowledges the support of the ANR as part of the "Investissements d'avenir" program, reference ANR-19-P3IA-0001 (PRAIRIE 3IA Institute). MS was supported by JST CREST Grant Number JPMJCR18A2.

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

## A  SUMMARY OF THE PROPOSED METHODS

Our proposed method can be summarized into two steps as shown in the following table.

Table 2: A summary of our proposed methods.

| Situation | Step 1: collect data | Step 2: estimate Bayes error |
|---|---|---|
| Clean soft labels | $\{c_i\}_{i=1}^n$ | $\hat{\beta} = \frac{1}{n} \sum_{i=1}^n \min\{c_i, 1 - c_i\}$ |
| Uncertainty labels | $\{c_i'\}_{i=1}^n$ | $\hat{\beta} = \frac{1}{n} \sum_{i=1}^n c_i'$ |
| Noisy soft labels | $\{(u_i, s_i)\}_{i=1}^n$ | $\hat{\beta}_{\text{Noise}} = \frac{1}{n} \left( \sum_{i=1}^{n_s^+}(1 - u_i^+) + \sum_{i=1}^{n_s^-} u_i^- \right)$ |
| Multiple hard labels | $\{y_{i,j}\}_{j=1}^m$ for $i \in [n]$ | $\tilde{\beta} = \frac{1}{n} \sum_i \min\{u_i, 1 - u_i\}$ where $u_i = \frac{1}{m} \sum_j \mathbb{1}[y_{i,j} = 1]$ |
| Positive confidence | $\{r_i\}_{i=1}^n$ and $\pi_+$ | $\hat{\beta}_{\text{Pconf}} = \pi_+ \left( 1 - \frac{1}{n_+} \sum_{i=1}^{n_+} \max\left( 0, 2 - \frac{1}{r_i} \right) \right)$ |

## B  RELATED WORK

**Bayes error estimation** has been a topic of interest for nearly half a century (Fukunaga & Hostetler, 1975). Several works have derived upper and lower bounds of the Bayes error and proposed ways to estimate those bounds. Various $f$-divergences, such as the Bhattacharyya distance (Fukunaga, 1990) or the Henze-Penrose divergence (Berisha et al., 2016; Sekeh et al., 2020), were studied. Other approaches include Noshad et al. (2019), which directly estimated the Bayes error with $f$-divergence representation instead of using a bound, and Theisen et al. (2021), which computed the Bayes error of generative models learned using normalizing flows (Kingma & Dhariwal, 2018). Michelucci et al. (2021) proposed a method to estimate the Bayes error and the best possible AUC for categorical features. More recently, Renggli et al. (2021) proposed a method to evaluate Bayes error estimators on real-world datasets, for which we usually do not have access to the true Bayes error.

All of these works that estimate the Bayes error are based on instance-label pairs. Not only do they require instances, but they also usually have a model, which may need careful tuning of hyperparameters to avoid overfitting. Li et al. (2021) proposed to estimate the maximal performance for regression problems, while we focus on the classification problem.

**Weakly supervised learning** tries to learn with a weaker type of supervision, e.g., semi-supervised learning (Chapelle et al., 2006; Sakai et al., 2017), positive-unlabeled learning (du Plessis et al., 2014), similar-unlabeled learning (Bao et al., 2018), unlabeled-unlabeled learning (Lu et al., 2019), and others (Zhou, 2018; Sugiyama et al., 2022).

On the other hand, the focus of most previous works on estimating the Bayes error is on ordinary classification problems with full supervision. To our knowledge, Bayes error estimation for weakly supervised data has not been studied. We discussed how our formulation introduced in this paper is flexible and can be extended to weakly supervised learning in Sec. 4.

**Overfitting to the test dataset** is one of the issues that concerns the machine learning community (Mania et al., 2019; Werpachowski et al., 2019; Kiela et al., 2021). Models can overfit to the test dataset by training on the test set, *explicitly* or *implicitly*. Barz & Denzler (2020) reported that 3% of the test data in CIFAR-10 has duplicates and near-duplicates in the training set. Lewis et al. (2020) studied popular open-domain question answering datasets and reported that 30% of test questions have a near-duplicate in the training data. These are examples of when we may explicitly train on the test set. Barz & Denzler (2020) and Lewis et al. (2020) showed how removing the duplicates will lead to substantially worse results.

Overfitting to the test set can also happen implicitly because research is conducted on the same test data that is public for many years (often with a fixed train/test partition), in contrast to how engineers in industry can sample fresh unseen data for evaluation. Recht et al. (2018) constructed CIFAR-10.1, an alternative unseen test set for CIFAR-10 classifiers and showed how deep learning models became worse with respect to the test error, with a 4% to 10% increase. Yadav & Bottou (2019) and Recht

et al. (2019) showed similar results for MNIST (LeCun et al., 1998) and ImageNet (Deng et al., 2009), respectively. With closer inspection, however, Recht et al. (2019) discussed how the accuracy improvement proportional to that in the original test set was observed in the new test set, for both CIFAR-10 and ImageNet. Roelofs et al. (2019) studied Kaggle competition datasets by comparing accuracies based on a public test set (which is used repeatedly by practitioners) and a private test set (which is used to determine the final rankings), but found little evidence of overfitting.

Constructing new test datasets is one way to examine the degree of overfitting to the test dataset. In Sec. 5, we discussed how estimating the Bayes error can be another way to detect overfitting to the test dataset. If models have lower test error than the estimated Bayes error, that is a strong signal for this type of overfitting. We compared the test error of recent classifiers with the estimated Bayes error in Sec. 5.

## C  PROOFS

### C.1  PROOF OF PROPOSITION 3.1

*Proof.*

$$\mathbb{E}_{\{\boldsymbol{x}_i\}_{i=1}^n \overset{\text{i.i.d.}}{\sim} p(\boldsymbol{x})}[\hat{\beta}] = \mathbb{E}_{\{\boldsymbol{x}_i\}_{i=1}^n \overset{\text{i.i.d.}}{\sim} p(\boldsymbol{x})}\left[\frac{1}{n}\sum_{i=1}^n [\min\{p(y=+1|\boldsymbol{x}_i), p(y=-1|\boldsymbol{x}_i)\}]\right] \quad (9)$$

$$= \frac{1}{n}\sum_{i=1}^n \mathbb{E}_{\boldsymbol{x}_i \sim p(\boldsymbol{x})}\left[\min\{p(y=+1|\boldsymbol{x}_i), p(y=-1|\boldsymbol{x}_i)\}\right] \quad (10)$$

$$= \frac{1}{n}\sum_{i=1}^n \beta \quad (11)$$

$$= \beta. \quad (12)$$

$\square$

### C.2  PROOF OF PROPOSITION 3.2

*Proof.* Since $0 \leq \min[p(y=+1|\boldsymbol{x}), p(y=-1|\boldsymbol{x})] \leq 0.5$, we can use the Hoeffding's inequality as follows:

$$P(|\hat{\beta} - \beta| \geq \epsilon) \leq 2\exp\left(-\frac{2n\epsilon^2}{\left(\frac{1}{2}\right)^2}\right) = 2\exp(-8n\epsilon^2). \quad (13)$$

Therefore, for any $\delta > 0$, with probability at least $1 - \delta$, we have,

$$|\hat{\beta} - \beta| \leq \sqrt{\frac{1}{8n}\log\frac{2}{\delta}}. \quad (14)$$

$\square$

## C.3 Proof that $\tilde{\beta}$ is a biased estimator

*Proof.*

$$\mathbb{E}_{\{c_i,\xi_i\}_{i=1}^n \overset{\text{i.i.d.}}{\sim} p(c,\xi)}[\tilde{\beta}] = \mathbb{E}_{\{c_i,\xi_i\}_{i=1}^n \overset{\text{i.i.d.}}{\sim} p(c,\xi)}\left[\frac{1}{n}\sum_{i=1}^n \left[\min\{u_i, 1-u_i\}\right]\right] \tag{15}$$

$$= \frac{1}{n}\sum_{i=1}^n \mathbb{E}_{(c_i,\xi_i)\sim p(c,\xi)}\left[\min\{u_i, 1-u_i\}\right] \tag{16}$$

$$\neq \frac{1}{n}\sum_{i=1}^n \mathbb{E}_{c_i\sim p(c)}\left[\min\{\mathbb{E}_{\xi_i\sim p(\xi|c_i)}[u_i], 1-\mathbb{E}_{\xi_i\sim p(\xi|c_i)}[u_i]\}\right] \tag{17}$$

$$= \frac{1}{n}\sum_{i=1}^n \mathbb{E}_{c_i\sim p(c)}\left[\min\{c_i, 1-c_i\}\right] \tag{18}$$

$$= \frac{1}{n}\sum_{i=1}^n \mathbb{E}_{\boldsymbol{x}_i\sim p(\boldsymbol{x})}\left[\min\{p(y=1|\boldsymbol{x}_i), p(y=-1|\boldsymbol{x}_i)\}\right] \tag{19}$$

$$= \frac{1}{n}\sum_{i=1}^n \beta = \beta \tag{20}$$

$\square$

## C.4 Proof of Proposition 3.6

For simplicity, suppose that we have $m$ labels for every sample $i$, and $u_i$ is the sample average of those labels:

$$u_i = \frac{1}{m}\sum_{j=1}^m \mathbb{1}[y_{i,j} = 1]. \tag{21}$$

Since $\mathbb{1}[y_{i,j} = 1] \in [0, 1]$, using Hoeffding's inequality, we get

$$\Pr\left[\bigcup_{i=1}^n \{|u_i - c_i| \geq \epsilon\} \mid c_i\right] \tag{22}$$

$$\leq \sum_{i=1}^n \Pr[|u_i - c_i| \geq \epsilon \mid c_i] \quad \text{(union bound)} \tag{23}$$

$$\leq 2n\exp\left(-\frac{2m\epsilon^2}{(1-0)^2}\right) \tag{24}$$

$$= 2n\exp\left(-2m\epsilon^2\right). \tag{25}$$

for any $\epsilon > 0$. In other words, for any $\delta' > 0$, with probability at least $1 - \delta'$, we have

$$|u_i - c_i| < \sqrt{\frac{1}{2m}\log\frac{2n}{\delta'}} =: \epsilon'(\delta') \tag{26}$$

for all $i = 1, \ldots, n$. Note that

$$\beta_i := \min\{c_i, 1-c_i\} \tag{27}$$

$$= \min\{\mathbb{E}[u_i \mid c_i], \mathbb{E}[1-u_i \mid c_i]\} \tag{28}$$

$$\geq \mathbb{E}[\min\{u_i, 1-u_i\} \mid c_i] \tag{29}$$

$$\geq \mathbb{E}[\tilde{\beta} \mid c_i]. \tag{30}$$

The conditional bias can be bounded as

$$\beta_i - \mathbb{E}[\tilde{\beta} \mid c_i] = \beta_i - \mathbb{E}\left[\frac{1}{n}\sum_{i=1}^{n}\min\{u_i, 1 - u_i\} \mid c_i\right] \tag{31}$$

$$= \frac{1}{n}\sum_{i=1}^{n}\mathbb{E}\left[\min\{c_i, 1 - c_i\} - \min\{u_i, 1 - u_i\} \mid c_i\right] \tag{32}$$

$$= \frac{1}{n}\sum_{i=1}^{n}\Pr[|u_i - c_i| \geq \epsilon'(\delta') \mid c_i]\mathbb{E}[\min\{c_i, 1 - c_i\} - \min\{u_i, 1 - u_i\} \mid c_i, |u_i - c_i| \geq \epsilon'(\delta')] \tag{33}$$

$$+ \frac{1}{n}\sum_{i=1}^{n}\Pr[|u_i - c_i| < \epsilon'(\delta') \mid c_i]\mathbb{E}[\min\{c_i, 1 - c_i\} - \min\{u_i, 1 - u_i\} \mid c_i, |u_i - c_i| < \epsilon'(\delta')] \tag{34}$$

$$\leq \frac{1}{n}\sum_{i=1}^{n}\Pr[|u_i - c_i| \geq \epsilon'(\delta) \mid c_i](1/2 - 0) \tag{35}$$

$$+ \frac{1}{n}\sum_{i=1}^{n}\Pr[|u_i - c_i| < \epsilon'(\delta') \mid c_i](\min\{c_i, 1 - c_i\} - \min\{c_i - \epsilon'(\delta'), 1 - c_i - \epsilon'(\delta')\}) \tag{36}$$

$$\leq \delta'/2 + \epsilon'(\delta'), \tag{37}$$

where the last line follows from Eq. (26. Eq. (36 holds because $\min\{c_i, 1-c_i\} \in [0, 1/2]$, $\min\{u_i, 1 - u_i\} \in [0, 1/2]$, and $|u_i - c_i| < \epsilon'(\delta')$ implies that $u_i > c_i - \epsilon'(\delta')$ and $1 - u_i > 1 - c_i - \epsilon'(\delta')$. Further taking the expectation on both sides, we get a bound on the bias:

$$\beta - \mathbb{E}[\tilde{\beta}] = \mathbb{E}[\beta_i - \mathbb{E}[\tilde{\beta} \mid c_i]] \tag{38}$$

$$\leq \delta'/2 + \epsilon'(\delta') \tag{39}$$

$$= \delta'/2 + \sqrt{\frac{1}{2m}\log\frac{2n}{\delta'}}. \tag{40}$$

Setting $\delta'$ to $m^{-1/2}$ yields

$$\beta - \mathbb{E}[\tilde{\beta}] \leq \frac{1}{2\sqrt{m}} + \sqrt{\frac{1}{2m}\log(2n\sqrt{m})} = \mathcal{O}\left(\sqrt{\frac{\log(2nm)}{m}}\right). \tag{41}$$

## C.5    PROOF OF PROPOSITION 3.4

*Proof.* $\hat{\beta}_{\text{Noise}}$ can be expressed as

$$\hat{\beta}_{\text{Noise}} = \frac{1}{n}\sum_{i=1}^{n}\{\mathbb{1}[c_i \geq 0.5](1 - u_i) + \mathbb{1}[c_i < 0.5]u_i\}. \tag{42}$$

Recall that $u_i = c_i + \xi_i$. Hence, we have

$$\mathbb{E}[\hat{\beta}_{\text{Noise}}] = \frac{1}{n}\sum_{i=1}^{n}\left(\mathbb{E}_{\{c_i,\xi_i\}_{i=1}^{n}\overset{\text{i.i.d.}}{\sim}p(r,\xi)}[\mathbb{1}[c_i \geq 0.5](1 - u_i)] + \mathbb{E}_{\{c_i,\xi_i\}_{i=1}^{n}\overset{\text{i.i.d.}}{\sim}p(r,\xi)}[\mathbb{1}[c_i < 0.5]u_i]\right) \tag{43}$$

$$= \frac{1}{n}\sum_{i=1}^{n}\left(\mathbb{E}_{\{c_i,\xi_i\}_{i=1}^{n}\overset{\text{i.i.d.}}{\sim}p(r,\xi)}[\mathbb{1}[c_i \geq 0.5](1 - c_i)] - \mathbb{E}_{\{c_i,\xi_i\}_{i=1}^{n}\overset{\text{i.i.d.}}{\sim}p(r,\xi)}[\mathbb{1}[c_i \geq 0.5]\xi_i]\right.$$

$$\left. + \mathbb{E}_{\{c_i,\xi_i\}_{i=1}^{n}\overset{\text{i.i.d.}}{\sim}p(r,\xi)}[\mathbb{1}[c_i < 0.5]c_i] + \mathbb{E}_{\{c_i,\xi_i\}_{i=1}^{n}\overset{\text{i.i.d.}}{\sim}p(r,\xi)}[\mathbb{1}[c_i < 0.5]\xi_i]\right). \tag{44}$$

The second term in Eq. (44) is,

$$\mathbb{E}_{\{c_i,\xi_i\}_{i=1}^n \overset{\text{i.i.d.}}{\sim} p(r,\xi)}[\mathbb{1}[c_i \geq 0.5]\xi_i] = \int\int \mathbb{1}[c_i \geq 0.5] \cdot \xi_i \cdot p(c_i,\xi_i)\mathrm{d}c_i\mathrm{d}\xi_i \tag{45}$$

$$= \int_{c_i \geq 0.5} \left(\int \xi_i p(\xi_i|c_i)\mathrm{d}\xi_i\right) p(c_i)\mathrm{d}c_i \tag{46}$$

$$= \int_{c_i \geq 0.5} \mathbb{E}[\xi_i|c_i]p(c_i)\mathrm{d}c_i \tag{47}$$

$$= 0. \tag{48}$$

Similarly, the fourth term in Eq. (44) becomes zero. Then,

$$\mathbb{E}[\hat{\beta}_{\text{Noise}}] = \frac{1}{n}\sum_{i=1}^n \left(\mathbb{E}_{c_i\sim p(c)}[\mathbb{1}[c_i \geq 0.5](1-c_i)] + \mathbb{E}_{c_i\sim p(c)}[\mathbb{1}[c_i < 0.5]c_i]\right) \tag{49}$$

$$= \frac{1}{n}\sum_{i=1}^n \mathbb{E}_{c_i\sim p(c)}\left[\min\{c_i, 1-c_i\}\right] \tag{50}$$

$$= \mathbb{E}_{c\sim p(c)}\left[\min\{c, 1-c\}\right] \tag{51}$$

$$= \mathbb{E}_{\boldsymbol{x}\sim p(\boldsymbol{x})}\left[\min\{p(y=1|\boldsymbol{x}), p(y=-1|\boldsymbol{x})\}\right] \tag{52}$$

$$= \beta. \tag{53}$$

$\square$

## C.6 PROOF OF PROPOSITION 3.5

*Proof.* $\hat{\beta}_{\text{Noise}}$ can be expressed as

$$\hat{\beta}_{\text{Noise}} = \frac{1}{n}\sum_{i=1}^n \check{b}_i, \tag{54}$$

where

$$\check{b}_i := \mathbb{1}[c_i \geq 0.5] \cdot (1-u_i) + \mathbb{1}[c_i < 0.5] \cdot u_i. \tag{55}$$

Since we can derive that $0 \leq \check{b}_i \leq 1$, we can again use the Hoeffding's inequality:

$$P(|\hat{\beta}_{\text{Noise}} - \beta| \geq \epsilon) \leq 2\exp\left(-\frac{2n\epsilon^2}{(1-0)^2}\right) = 2\exp\left(-2n\epsilon^2\right). \tag{56}$$

Therefore, for any $\delta > 0$, with probability at least $1 - \delta$, we have,

$$|\hat{\beta}_{\text{Noise}} - \beta| \leq \sqrt{\frac{1}{2n}\log\frac{2}{\delta}}. \tag{57}$$

$\square$

## C.7 PROOF OF THEOREM 4.1

*Proof.*

$$\beta = \mathbb{E}_{p(\boldsymbol{x})}[\min\{p(y=+1|\boldsymbol{x}), p(y=-1|\boldsymbol{x})\}] \tag{58}$$

$$= \int \min\{p(y=+1|\boldsymbol{x}), p(y=-1|\boldsymbol{x})\}p(\boldsymbol{x})\mathrm{d}\boldsymbol{x} \tag{59}$$

$$= \int \min\{p(y=+1|\boldsymbol{x}), p(y=-1|\boldsymbol{x})\}\frac{\pi_+ p(\boldsymbol{x}|y=+1)}{p(y=+1|\boldsymbol{x})}\mathrm{d}\boldsymbol{x} \tag{60}$$

$$= \int \min\{p(y=+1|\boldsymbol{x}), 1-p(y=+1|\boldsymbol{x})\}\frac{\pi_+ p(\boldsymbol{x}|y=+1)}{p(y=+1|\boldsymbol{x})}\mathrm{d}\boldsymbol{x} \tag{61}$$

$$= \int \min\left\{1, \frac{1-p(y=+1|\boldsymbol{x})}{p(y=+1|\boldsymbol{x})}\right\}\pi_+ p(\boldsymbol{x}|y=+1)\mathrm{d}\boldsymbol{x} \tag{62}$$

$$= \pi_+ \mathbb{E}_+\left[\min\left\{1, \frac{1-r(\boldsymbol{x})}{r(\boldsymbol{x})}\right\}\right] \tag{63}$$

$$= \pi_+\left(1 + \mathbb{E}_+\left[\min\left\{0, \frac{1}{r(\boldsymbol{x})}-2\right\}\right]\right) \tag{64}$$

$\square$

## C.8 PROOF OF PROPOSITION 4.2

*Proof.*

$$\mathbb{E}_{\{\boldsymbol{x}_i\}_{i=1}^n \overset{\text{i.i.d.}}{\sim} p(\boldsymbol{x}|y=+1)}[\hat{\beta}_{\text{Pconf}}] = \pi_+ - \frac{\pi_+}{n}\sum_{i=1}^n \mathbb{E}_+\left[\max\left\{0, 2-\frac{1}{r(\boldsymbol{x}_i)}\right\}\right] \tag{65}$$

$$= \pi_+\left(1 - \mathbb{E}_+\left[\max\left\{0, 2-\frac{1}{r(\boldsymbol{x})}\right\}\right]\right) \tag{66}$$

$$= \pi_+\left(1 + \mathbb{E}_+\left[\min\left\{0, \frac{1}{r(\boldsymbol{x})}-2\right\}\right]\right) \tag{67}$$

$$= \pi_+\left(1 + \left(\frac{\beta}{\pi_+}-1\right)\right) \tag{68}$$

$$= \beta \tag{69}$$

In the third equality, we used $\max\{a,b\} = -\min\{-a,-b\}$. In the fourth equality, we used $\mathbb{E}_+\left[\min\left\{0, \frac{1}{r(\boldsymbol{x})}-2\right\}\right] = \frac{\beta}{\pi_+}-1$ which can be derived from Eq. (64). $\square$

## C.9 PROOF OF PROPOSITION 4.3

We can rewrite $\hat{\beta}_{\text{Pconf}}$ as

$$\hat{\beta}_{\text{Pconf}} = \frac{1}{n_+}\sum_{i=1}^{n_+}\left(\pi_+ - \pi_+ \max\left\{0, 2-\frac{1}{r(\boldsymbol{x}_i)}\right\}\right) \tag{70}$$

Although $1/r(\boldsymbol{x}_i)$ is not bounded, since $0 \le \pi_+ - \pi_+ \max\left(0, 2-\frac{1}{r(\boldsymbol{x}_i)}\right) \le \pi_+$, we can use the Hoeffding's inequality to derive,

$$P(|\hat{\beta}_{\text{Pconf}} - \beta| \ge \epsilon) \le 2\exp\left(\frac{-2n\epsilon^2}{\pi_+^2}\right) \tag{71}$$

Therefore, for any $\delta > 0$, with probability at least $1 - \delta$ we have,

$$|\hat{\beta}_{\text{Pconf}} - \beta| \le \sqrt{\frac{\pi_+^2}{2n}\log\frac{2}{\delta}}. \tag{72}$$

## D STOCHASTIC BIG-O NOTATION

Suppose $\{X_n\}$ ($n = 1, 2, \ldots$) is a sequence of random variables and $\{a_n\}$ ($n = 1, 2, \ldots$) is a sequence of positive numbers, respectively. We say $X_n = O_p(a_n)$ when for all $\epsilon > 0$, there exists $C \in \mathbb{R}$ and finite $N$ such that for all $n > N$, $P(\frac{|X_n|}{a_n} > C) < \epsilon$.

## E DATA COLLECTION

We chose the Fashion-MNIST dataset (Xiao et al., 2017) as a more challenging dataset. This is a 10 class classification dataset with small fashion images with categories such as dress, bag, and sneaker. [6] We used Amazon Mechanical Turk (Buhrmester et al., 2016) to collect multiple annotations per each image in the test dataset with 10,000 images. We designed the task so that the worker is instructed to annotate 10 randomly chosen fashion images, with a multiple choice of 10 labels. We showed several correct images for each of categories in the beginning of the task, so that the worker can have a good understanding of each labels. For each task with 10 image annotations, we paid $0.06 or $0.08 to the worker. We removed the duplicates when the same worker provide a label for the same image more than once. We further removed the annotation when the worker provided an answer within 30 seconds, which we considered as too short, e.g., when we performed the task, it took us 45 seconds when we skipped reading the instructions. After removing the shorter ones, the mean and median answering time was about 147 and 78 seconds, respectively. Finally, we ended up with around 67 annotations per image. The least number was 53 and the largest number was 79 annotations. By following the naming convention of CIFAR-10H, we call this dataset as Fashion-MNIST-H.

## F FULL RESULTS FOR SEC. 5.1 AND SEC. 5.2

In Sec. 5.1 and Sec. 5.2, we only showed two setups for each set of experiments. In this section, we show all four setups.

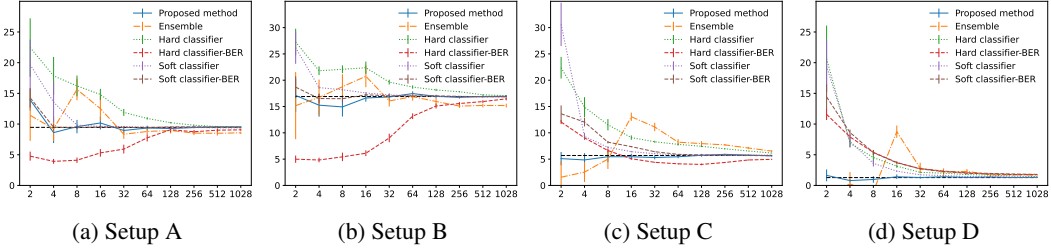

|(a) Setup A|(b) Setup B|(c) Setup C|(d) Setup D|

Figure 5: Comparison of the proposed method (blue) with the true Bayes error (black), a recent Bayes error estimator "Ensemble" (orange) (Noshad et al., 2019), and the test error of a multilayer perceptron (hard labels: green, soft labels: purple). We also use the classifier's probabilistic output and plug it into the Bayes error equation in Eq. (3), shown as "Classifier-BER" (hard labels: red, soft labels: brown). The vertical/horizontal axis is the test error or estimated Bayes error/# of training samples per class, respectively. Instance-free data was used for Bayes error estimation, while fully labelled data was used for training classifiers. We plot the mean and standard error for 10 trials.

---

[6]Other popular datasets such as CIFAR-100 (Krizhevsky, 2009) and ImageNet (Deng et al., 2009) are problematic because they are known to be essentially multi-label classification datasets rather than single-label (Shankar et al., 2020; Beyer et al., 2020; Yun et al., 2021; Wei et al., 2022). Since the Bayes error is defined for the single-labeled case, this means we have to define a similar but different and novel concept that corresponds to the lowest error for multi-label classification if we want to estimate the best possible performance on datasets such as CIFAR-100 and ImageNet. This is definitely an interesting direction to pursue in the future, but is out of the scope of the current paper since we are focusing on the Bayes error.

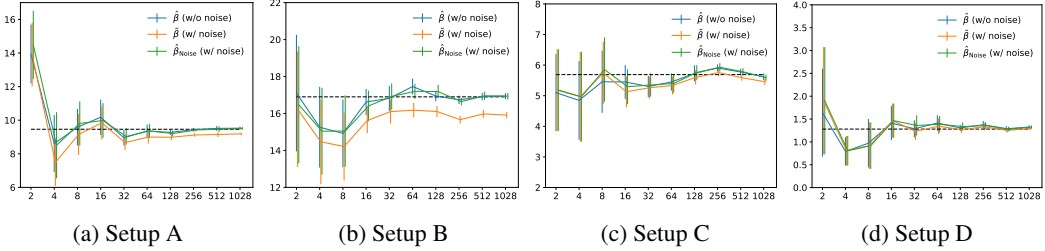

Figure 6: Bayes error with and without noise on the labels, with mean and standard error for 10 trials. The vertical/horizontal axis is the estimated Bayes error/the number of samples. With noisy labels, the estimation tends to become inaccurate with a larger standard error. With more samples, the difference between clean labels becomes smaller, but a small bias remains. With our modified estimator, the bias vanishes.

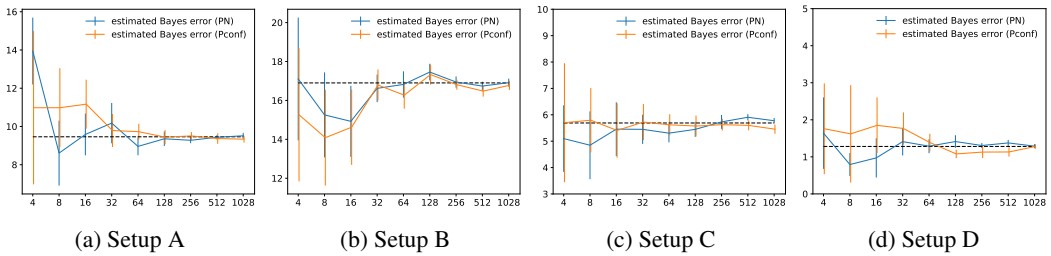

Figure 7: Estimated Bayes error with PN (positive and negative) and Pconf (positive-confidence). The vertical/horizontal axis is the estimated Bayes error/number of samples. We plot the mean and standard error for 10 trials. The difference between the two becomes smaller as we increase the number of training samples.

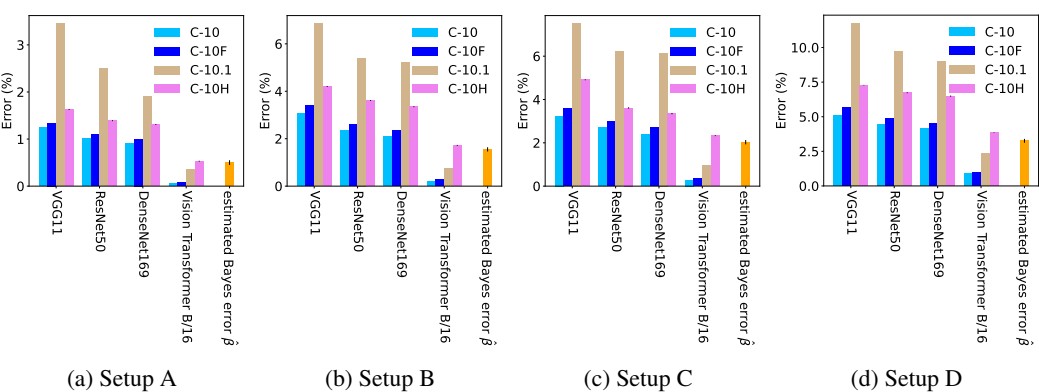

Figure 8: Comparing the estimated Bayes error with four classifiers and four setups (Figures 8a, 8b, 8c, and 8d). Although it seems that ViT B/16 has surpassed the Bayes error for all four setups for CIFAR-10 (C10), ciFAIR-10 (C10F), and CIFAR-10.1 (C10.1), it is still comparable or larger than the Bayes error for CIFAR-10H (C10H). In Sec. 5.2, we discuss how this has to be investigated carefully. The black error bar for the Bayes error is the $95\%$ confidence interval. The black error bar for the C10H is the standard error for sampling test labels 20 times.

