# OpenReview forum: "Is the Performance of My Deep Network Too Good to Be True? A Direct Approach to Estimating the Bayes Error in Binary Classification"
_ICLR.cc/2023/Conference — ICLR 2023 notable top 5%_

### Official Review · Reviewer_bzQr · 2022-10-20

**Confidence:** 3
**Correctness:** 3
**Technical Novelty And Significance:** 3
**Empirical Novelty And Significance:** 4
**Recommendation:** 8

**Clarity, Quality, Novelty And Reproducibility:**

## Clarity

Other than minor points raised above, the text is clear.

## Quality

The overall quality is good.

## Novelty

To my knowledge, the findings presented are sufficiently novel.

**Strength And Weaknesses:**

## Strengths

- The authors provide many rigorous proofs to support their claims.
- The empirical results provide further support, showing bias where expected, etc.


## Weaknesses

- Theorem 2.2 is introduced from previous work without much context around why it is useful or important for this work. It would be helpful to the reader to provide some context of where and why it will come into play later in the text.

- Until section 3, there is no mention of where the soft labels come from. It seems that the soft labels would have to be soft labels that are given from the true distribution of the data, which it seems would be impossible to attain in most realistic circumstances. Although, this is explained later, I think it would reduce reader confusion by at least alluding to this fact and that it will be dealt with later.

- Section 3, paragraph 3: Why can uncertainty labels not recover $p(y = +1 | \mathbf{x})$ from $\min_{y \in \pm 1}p(y | \mathbf{x})$? If one has access to one, doesn't it guarantee we know the other? If $\min_{y \in \pm 1}p(y | \mathbf{x}) = 0.1$, then isn't $p(y = +1 | \mathbf{x}) = 0.9$ If this is a subtle misunderstanding about the meaning of soft labels and uncertainty labels introduced in Section 1, then I think there needs to be more time devoted to explaining those concepts in detail.

- Theorem 3.3 seems to show that if one has a noisy corruption of the true class of an instance, then $\hat{\beta}$ is an unbiased estimate of the Bayes error. Later on in page 4, the authors state "In practice, one idea is to ask many labellers for each instance and then use the histogram of the hard labels as a noisy soft label." Therefore wouldn't this just produce an unbiased estimate of the Bayes error of the aggregated labellers, which is not in fact an unbiased estimate of the true Bayes error? For example, the estimated Bayes error of a ResNet on CIFAR-10H is higher than that of the human annotated $\hat{\beta}$ in figure 4. This might be an obvious point, but I still think it should be stated clearly somewhere in the text.

- It would be interesting to see the other baseline performances on something other than the toy dataset. For example, ensembles tend to not show much diversity in decision boundaries when considering toy datasets, and therefore often do not outperform simple MLP baselines by much. This changes when the dimensionality of the data increases. It would therefore be interesting to see what happens with the Bayes error of an ensemble of at least one of the network baselines in figure 4.


## Minor

- Page 4: reference error, "which we use later on in Sec. ??"
- Page 8: reference error, "(App. ??)"

**Summary Of The Paper:**

The authors propose a method of estimating the Bayes' error of a class of models for binary classification. They compare the estimation with datasets with multiple human annotated labels to compare the estimated Bayes error with an empirical estimate of the Bayes error.

**Summary Of The Review:**

Overall, I feel positive about the results presented in the text. I raised a number of points above, and I look forward to hearing the responses from the authors regarding the points raised.

---

> ### Author Response · Authors · 2022-11-16
> **Reply**
>
> We thank Reviewer bzQr for reviewing our paper and for the insightful comments. We hope our answers to the 5 questions will address the concerns and clarify the contributions of the paper.
>
> **Q1)** Theorem 2.2 is introduced from previous work without much context around why it is useful or important for this work. It would be helpful to the reader to provide some context of where and why it will come into play later in the text.
>
> **A1)** Thank you for the suggestion! We updated some sentences in Section 2 in our updated paper to provide some context of Theorem 2.2.
>
> **Q2)** Until section 3, there is no mention of where the soft labels come from. It seems that the soft labels would have to be soft labels that are given from the true distribution of the data, which it seems would be impossible to attain in most realistic circumstances. Although, this is explained later, I think it would reduce reader confusion by at least alluding to this fact and that it will be dealt with later.
>
> **A2)** Thank you! We updated footnote 2 so that readers will not be confused.  (Unfortunately we had to use the footnote due to the tight space constraint.)
>
> **Q3)** Section 3, paragraph 3: Why can uncertainty labels not recover $p(y=+1|x)$ from $\\min\_{y \\in \\pm 1} p(y|x)$? If one has access to one, doesn't it guarantee we know the other? If $\\min\_{y \\in \\pm 1} p(y|x) = 0.1$, then isn't $p(y=+1|x)=0.9$? If this is a subtle misunderstanding about the meaning of soft labels and uncertainty labels introduced in Section 1, then I think there needs to be more time devoted to explaining those concepts in detail.
>
> **A3)** Even if we know that $\\min\_{y \\in \\{\\pm 1\\}} p(y|x) = 0.1$, we cannot identify which one of the following holds: $p(y=+1|x) = 0.9$ or $p(y=+1|x)=0.1$, because they both will mean $\\min\_{y \\in \\{\\pm 1\\}} p(y|x) = 0.1$. We added a footnote in our updated paper to explain this in detail.
>
> **Q4)** Theorem 3.3 seems to show that if one has a noisy corruption of the true class of an instance, then $\\tilde{\\beta}$  is an unbiased estimate of the Bayes error. Later on in page 4, the authors state "In practice, one idea is to ask many labellers for each instance and then use the histogram of the hard labels as a noisy soft label." Therefore wouldn't this just produce an unbiased estimate of the Bayes error of the aggregated labellers, which is not in fact an unbiased estimate of the true Bayes error? For example, the estimated Bayes error of a ResNet on CIFAR-10H is higher than that of the human annotated $\\hat{\\beta}$  in figure 4. This might be an obvious point, but I still think it should be stated clearly somewhere in the text.
>
> **A4)** We claim the opposite in Theorem 3.3: $\\tilde{\\beta}$ is a *biased* estimator. If we "ask many labellers for each instance and then use the histogram of the hard labels as a noisy soft label", then we will have to use the biased $\\tilde{\\beta}$, so we think the reviewer is correct that it will be biased, but in Prop 3.6 we show that it is asymptotically unbiased. We will clarify this in the corresponding sections.
>
> **Q5)** It would be interesting to see the other baseline performances on something other than the toy dataset. For example, ensembles tend to not show much diversity in decision boundaries when considering toy datasets, and therefore often do not outperform simple MLP baselines by much. This changes when the dimensionality of the data increases. It would therefore be interesting to see what happens with the Bayes error of an ensemble of at least one of the network baselines in figure 4.
>
> **A5)** In Figure 1, "Ensemble" is not an ensemble of many classifiers, but is a Bayes error estimator proposed by Noshad et al. 2019, and it does not have a decision boundary that classifiers or ensembles of classifiers have. We conjecture that since the ensemble method in Noshad et al. 2019 is based on an epsilon-ball estimation, it will not work so well with high-dimensional datasets such as CIFAR-10 in Figure 4, unless we combine it with dimensionality reduction and use a low dimensional representation of the data.

---

> > ### Comment · Reviewer_bzQr · 2022-11-24
> > **Thank you for the responses.**
> >
> > Thank you for the responses. My questions have been answered and I find this work interesting, so I will raise my score accordingly.

---

> > > ### Author Response · Authors · 2022-11-25
> > > **Reply**
> > >
> > > Thank you for reading our rebuttal and for updating the review!

---

### Official Review · Reviewer_5ZHz · 2022-10-24

**Confidence:** 2
**Correctness:** 3
**Technical Novelty And Significance:** 3
**Empirical Novelty And Significance:** 2
**Recommendation:** 5

**Clarity, Quality, Novelty And Reproducibility:**

The presentation of the draft is clear to me. The proposed estimator is simple and straightforward. I am not sure if it has been proposed before or not.


**Strength And Weaknesses:**

Strength: The proposed estimator is simple to implement. Also it is nice that the authors show some guarantee of the proposed estimator.


Weakness: I feel the proposed estimator does not really resolve the core of the Bayes error estimation. For the theory part of the paper, e.g., Theorem 4.1 and Proposition 4.2 and 4.3, the assumption is r_i= p(y=+1|x_i) is known. But to me this is actually the core and hard part of Bayes error estimation, since given r_i one can easily get the Bayes estimator.


In the empirical section of the paper, p(y=+1|x_i) is estimated using deep models, but as we all know from the uncertainty estimation literatures, deep models are not doing a good job calibrating their output scores to p(y=+1|x_i), and their outputs are usually highly screwed. As a result I am not quite sure how I should trust the estimated Bayes error.


Experiments are also kind of lacking. I only see two real-world datasets used. To fully show empirically the proposed estimator work I would like to see more numbers on other datasets.


**Summary Of The Paper:**

This paper proposes a simple estimator for Bayes error in binary classification. The authors are able to show several properties of the proposed estimator including unbiasedness and some convergence guarantee. Experiments on synthetic datasets, CIFAR-10, and Fashion-MNIST show the effectiveness of the proposed estimator.


**Summary Of The Review:**

Overall I feel the work is an interesting try and it does show some promising preliminary results. But as I stated in the weakness section I do have some serious concerns. The assumption that p(y=+1|x_i) is known is so strong that the problem is not interesting any more in theory. In application, the work does not propose convincing ways of estimating p(y=+1|x_i) either. And the experiments are not extensive either.

---

> ### Author Response · Authors · 2022-11-16
> **Reply**
>
> We thank Reviewer 5ZHz for reviewing our paper and for the insightful comments. We hope our answers to the 3 questions will address the concerns and clarify the contributions of the paper.
>
> **Q1)** I feel the proposed estimator does not really resolve the core of the Bayes error estimation. For the theory part of the paper, e.g., Theorem 4.1 and Proposition 4.2 and 4.3, the assumption is $r_i= p(y=+1|x_i)$ is known. But to me this is actually the core and hard part of Bayes error estimation, since given r_i one can easily get the Bayes estimator.
>
> **A1)** We start our discussions by assuming that we can collect $c_i=p(y=+1|x_i)$ in the usual case (or $r_i$ in Pconf case), but we do not assume full knowledge of $p(y=+1|x)$ for all $x$, but only for some finite instances. That being said, we relax this assumption in several ways. We first discuss how we can use a noisy version instead with unbiasedness (Proposition 3.4) and consistency (Proposition 3.5). We also demonstrate that the hard labels in the standard classification case are strictly special cases of the noisy soft labels (end of Sec. 3), implying our zero-mean noise assumption is realistic to some extent. Finally, we discuss how we can collect multiple hard labels instead, and show that this will lead to an asymptotically unbiased estimator in Proposition 3.6.
>
> We believe our main contributions are about Bayes error estimation for the usual classification setup in Section 3, although the contributions in the weakly-supervised setup in Section 4 (with Thm 4.1, Props 4.2 & 4.3) are also a nice byproduct. We are looking forward to working on relaxing the assumptions for the weakly-supervised version in the future.
>
>
> **Q2)** In the empirical section of the paper, $p(y=+1|x_i)$ is estimated using deep models, but as we all know from the uncertainty estimation literatures, deep models are not doing a good job calibrating their output scores to $p(y=+1|x_i)$, and their outputs are usually highly screwed. As a result I am not quite sure how I should trust the estimated Bayes error. In application, the work does not propose convincing ways of estimating $p(y=+1|x_i)$.
>
> **A2)** We do *not* take this approach. We use soft labels (or its variants) based on human annotations instead of utilizing the classifier’s probabilistic outputs. The probabilistic outputs of classifiers are only used as a baseline, and one reason this problematic approach underperforms our proposed approach is definitely the calibration issue of neural networks. We can see this from Figure 1 (c) and (d), where the red (Hard classifier-BER) tends to severely underestimate the true Bayes error. This is basically because the neural network becomes overconfident, then $\\min \\{\\hat{p}(y=+1|x), \\hat{p}(y=-1|x)\\}$ becomes smaller, and finally, the estimated Bayes error becomes smaller. We will clarify this part in our paper.
>
>
> **Q3)** Experiments are also kind of lacking. I only see two real-world datasets used. To fully show empirically the proposed estimator work I would like to see more numbers on other datasets.
>
> **A3)** We would like to emphasize that the goal of the experiments that compare with recent classifiers such as ViT is *not* to show that our proposed estimator works, but rather an application of Bayes error estimation. This is because we usually do not know the underlying true Bayes error rate for real-world datasets, so even if we add more real-world datasets, that is not necessarily helpful to gain confidence that the estimation is accurate.  Hence, it is important to investigate how accurate our proposed method can be with datasets where the Bayes error can be precisely derived before we work on these applications. This is why we carefully demonstrate a lot of different distributions with simulated data in Sec. 5.1 and compare with the true Bayes error. From this perspective, we believe that having 2 real-world datasets already provides enough additional insights to the reader given that this is an application rather than verification.
>
> However, we agree that having more real-world datasets may be helpful and will make the paper more interesting. If the reviewer has any suggestions for the 3rd one, we will be happy to work on it. To our knowledge, it seems that there is only 1 existing dataset with multiple (>dozen) annotations per instance, which is CIFAR-10H used in Sec. 5.2. Therefore, we collected around 70 annotations per image for the Fashion-MNIST dataset in Sec. 5.3, so that we can apply our method to a 2nd real-world dataset. We also considered others such as CIFAR-100 and ImageNet, but noticed these are not appropriate for the reasons we discussed in Sec. 5.3.

---

### Official Review · Reviewer_YXWn · 2022-10-25

**Confidence:** 3
**Correctness:** 4
**Technical Novelty And Significance:** 3
**Empirical Novelty And Significance:** 3
**Recommendation:** 8

**Clarity, Quality, Novelty And Reproducibility:**

The presentation is clear overall - some references need to be fix (see the above paragraphs). The novelty and quality are good.

**Strength And Weaknesses:**

Strengths:
- The concept is conceptually simple and well-motivated.
- The presentation is clear overall.
- The experiments involve ``real-world'' scenes which may bring practical interest
- The proposed estimators works (except on ViT and some datasets) better than other trained models in estimating the Bayes error.

Questions:
- Some reference are missing. For example, page 4 has a ''Sec. ??'' and page 8 has a ''App. ??''.
- One paper [1] might be worthy to discuss (and maybe compare) but seems to be missing. It is also a model-agnostic algorithm for estimating the Bayes error, although they require the access to instances.
- I might be wrong but since human labeler are also imperfect, why is it not possible that ViT can outperform human in providing reliable confidence?

[1] A Model-Agnostic Algorithm for Bayes Error Determination in Binary Classification

**Summary Of The Paper:**

This paper proposes an instance-free and model-free Bayes error estimation method for weakly supervised data. It leverages the fact that the expectation of the (perfect) predicted probability is the unbiased estimator of the Bayes error. Further, in a more practical and general case where the noises exist, the estimator is asymptotically unbiased. The authors also studied one case where only positive data and their confidence are available, and they show the error is also bounded.

**Summary Of The Review:**

The paper provides an interesting method for estimating the Bayes error that is conceptually simple but theroetically grounded. I believe it would be beneficial to the community and inspire future research.

---

> ### Author Response · Authors · 2022-11-16
> **Reply**
>
> We thank Reviewer YXWn for reviewing our paper and for the insightful comments. We hope our answers to the 3 questions will address the concerns and clarify the contributions of the paper.
>
> **Q1)** Some reference are missing. For example, page 4 has a ''Sec. ??'' and page 8 has a ''App. ??''.
>
> Thank you for pointing this out! We fixed these in our updated paper.
>
> **Q2)** One paper ( [1] A Model-Agnostic Algorithm for Bayes Error Determination in Binary Classification) might be worthy to discuss (and maybe compare) but seems to be missing. It is also a model-agnostic algorithm for estimating the Bayes error, although they require the access to instances.
>
> Thank you for the suggestion! This paper [1] is based on a setting where features must be categorical and can take only a finite set of values, while our method does not have restrictions on the features, i.e., we can use our proposed method for both categorical and continuous features. Since all our experiments are based on features that are continuous, we cannot add [1] as a baseline in the experiments. We added discussions about this paper in Section 1 and Appendix A.
>
> **Q3)** I might be wrong but since human labeler are also imperfect, why is it not possible that ViT can outperform human in providing reliable confidence?
>
> It is conceptually impossible for ViT to outperform the estimated Bayes error based on human labelers when there is no distribution shift, i.e. $p(y|x) = q(y|x)$, where the classifier’s training data $(x, y)$ is sampled from $p(x, y) = p(y|x)p(x)$ and $q(y|x)$ is the soft label. On the other hand, if there are some kind of shifts, meaning $p(y|x) \neq q(y|x)$, we believe, as the reviewer points out, ViT’s test error may be smaller than the estimated Bayes error. We discuss this to some extent on page 8.
>
> Since having no shift means that human labelers’ $q(y|x)$ is the same as the underlying $p(y|x)$, it may seem unrealistic at first glance. However in real-world classification problems, there are many situations where, in fact, we are interested in $q(y|x)$ itself. This happens whenever we regard the human hard labels as the ground truth for the classification task, and the various “human noise” that may be incorporated into the label decision is no-longer considered as noise that leads to distribution shift, because we use that for our test labels. In other words, we are implicitly hoping to predict the human labels. Unfortunately, this is not always the case. The noise in human labels becomes an issue when we have test labels that are collected with a different labeling mechanism, e.g., in ImageNet, there are some discussions about this issue such as the last paragraph of Section 1 in Shankar et al. (ICML, 2020) and Section “Computing Human Accuracy” in Karpathy (2014). On the other hand, there are many situations where we use the human labels as the ground truth. Francis (2020) gives an example of crater detection in Mars from satellite images. Unless we have the full knowledge of all previous meteorites and volcanic eruptions in the history of Mars, we may have to resort to asking (expert) human annotators to detect the craters from the satellite images and use that as the ground truth. Sasada et al. (Leukemia Research 2018) provides a similar example in cell image classification, and they use human (medical technologist) labels as the ground truth.
>
> We thank the reviewer for the question, and we would like to add this discussion in our final version.
>
> References:
> - Vaishaal Shankar, Rebecca Roelofs, Horia Mania, Alex Fang, Benjamin Recht, Ludwig Schmidt. Evaluating Machine Accuracy on ImageNet. ICML 2020.
> - Andrej Karpathy. What I learned from competing against a ConvNet on ImageNet. http://karpathy.github.io/2014/09/02/what-i-learned-from-competing-against-a-convnet-on-imagenet/
> - Keiko Sasada, Noriko Yamamoto, Hiroki Masuda, Yoko Tanaka, Ayako Ishihara, Yasushi Takamatsu, Yutaka Yatomi, Waichiro Katsuda, Issei Sato, Hirotaka Matsui, on behalf of the Kyushu regional department of the Japanese Society of Laboratory Hematology. Inter-observer variance and the need for standardization in the morphological classification of myelodysplastic syndrome. Leukemia Research, 69, pp54-59, 2018.
> - Francis, A., Brown, J., Cameron, T., Crawford Clarke, R., Dodd, R., Hurdle, J., ... & Muller, J. P. (2020). A multi-annotator survey of sub-km craters on Mars. Data, 5(3), 70.

---

> > ### Comment · Reviewer_YXWn · 2022-11-24
> > **Post rebuttal discussion**
> >
> > Thanks to the authors for the efforts in responding to my questions. I can see the difference between the related work I pointed to now. All my concerns have been addressed.

---

> > > ### Author Response · Authors · 2022-11-25
> > > **Reply**
> > >
> > > Thank you for reading our rebuttal!

---

### Official Review · Reviewer_Y3qe · 2022-10-25

**Confidence:** 2
**Correctness:** 4
**Technical Novelty And Significance:** 4
**Empirical Novelty And Significance:** 4
**Recommendation:** 8

**Clarity, Quality, Novelty And Reproducibility:**

This paper is well presented. But it would be favored if a flowchart of the whole method can be provided.

**Strength And Weaknesses:**

Strengths:

* This paper is well-written, and there are many mathematical formulations to formally illustrate the logic and soundness of the proposed method.
* The experimental results seem promising.

Weaknesses:

* The proposed method seems simple. However, it is not clear in practice how a practitioner implements the whole method.

* The method is supposed to be instance free. However, I still see that real instances would be desired in experiments.

**Summary Of The Paper:**

To examine the intrinsic soundness of the classifier (maybe formulated as a deep neural network), this paper proposes a new method to estimate the limit Bayes error, where we just take the mean of the labels that show the uncertainty of the classes. The proposed method is claimed to be model-free and instance-free, and experimental results show its effectiveness.

**Summary Of The Review:**

Please see the above analysis.

---

> ### Author Response · Authors · 2022-11-16
> **Reply**
>
> We thank Reviewer Y3qe for reviewing our paper and for the insightful comments. We hope our answers to the 2 questions will address the concerns and clarify the contributions of the paper.
>
> **Q1)** The proposed method seems simple. However, it is not clear in practice how a practitioner implements the whole method. / This paper is well presented. But it would be favored if a flowchart of the whole method can be provided.
>
> Thank you for the suggestion! Here is our initial flowchart (more like a table because there are actually only 2 steps). We will add this to our paper.
>
> | Situation | Step 1: collect soft labels | Step 2: estimate Bayes error|
> | ---- | ---- | ----|
> | Clean soft labels | collect $\\{c_i\\}^n_{i=1}$ | $\\hat{\\beta} = \\frac{1}{n} \\sum \\text{min} \\{ c_i, 1-c_i \\}$ |
> |Noisy soft labels | collect $\\{u_i\\}\_{i=1}^n$ and $\\{s_i\\}\_{i=1}^n$ | $\\hat{\\beta}\_{\\mathrm{Noise}} = \\frac{1}{n} \\left(\\sum_{i=1}^{n_s^+} \\left(1-u^+\_i\\right) + \\sum_{i=1}^{n\_s^-} u^-\_i \\right)$
> | Multiple hard labels | collect $\\{y_i\\}\_{i=1}^n$ and derive $u_i = \frac{1}{m} \sum_{j=1}^m 1[y_i = 1]$  | $\tilde{\beta} = \frac{1}{n} \sum^n\_{i=1} \text{min} \\{u_i, 1-u_i\\}$ |
> | Positive confidence | collect $\\{r_i\\}\_{i=1}^n$ and $\pi_+$ | $\\hat{\\beta}\_{\\mathrm{Pconf}} = \\pi_+ \\left( 1 - \\frac{1}{n\_+} \\sum^{n\_+}\_{i=1} \\text{max} \\left(0, 2 - \\frac{1}{r\_i}\right) \\right)$ |
>
> **Q2)** The method is supposed to be instance free. However, I still see that real instances would be desired in experiments.
>
> The proposed method does not use instances even in the experiments. Other Bayes error estimation baselines require instances in the experiments so we provide instances when we use the baselines. We are trying to make it advantageous for others so that we can clearly see the superiority of our method.

---

### Decision · Program_Chairs · 2023-01-20

**Decision:**

Accept: notable-top-5%

**Justification For Why Not Higher Score:**

N/A

**Justification For Why Not Lower Score:**

Overall, this paper (1) studies a novel problem with practical motivation; (2) brings in well-grounded theory insights; (3) is extensively tested on real data, including an interesting acceptance/rejection binary classification task for ICLR papers; (4) is well written.

The current average score is 8-8-8-5, while the score 5 should be down-weighted since the reviewer was unfortunately not responsive. Hence, I consider this paper top-notch quality among all submissions within my batch, and would like to recommend oral.

**Metareview: Summary, Strengths And Weaknesses:**

This paper proposes an instance-free, hyperparameter-free and model-free Bayes error estimation method for weakly supervised data. The authors point out that the expectation of the (perfect) predicted probability is the unbiased estimator of the Bayes error. Further, in a more practical case where the noise exists, the estimator is still asymptotically unbiased. This method is conceptually simple, well-motivated, and practically tested in a partial label setting where positive data and their confidences are available. Writing is clear and logical.

The experiments involve ``real-world'' scenes which may bring practical interest. During rebuttal, the authors provide another set of real-world experiments (acceptance/rejection binary classification task for ICLR papers) to further justify their method's robustness. I applaud the authors for being very thorough, solid, and convincing in their rebuttal process.

After rebuttal, three out of four reviewers unanimously found this work very interesting and rate it highly as 8. The only negative reviewer 5ZHz (rating as 5) didn't respond to either the authors' rebuttal (which seems to have addressed all raised questions and confusions) or AC's discussion request, despite multiple reminders: hence his/her rating has been down-weighted.



**Note From Pc:**

if the above contains the word "oral" or "spotlight" please see: "oral" presentation means -> notable-top-5% and "spotlight" means -> notable-top-25%. As stated in our emails, we are disassociating presentation type from AC recommendations